# Things Always Come in Three: Non-Invasive Investigations of Alexander and Roxane's Wedding Room in Villa Farnesina

**Manuela Vagnini** [1,*]**, Chiara Anselmi** [2,*] **, Michela Azzarelli** [1] **and Antonio Sgamellotti** [3,4]

1   Laboratorio di Diagnostica Per i Beni Culturali, Piazza Campello 2, 06049 Spoleto, Italy; m.azzarelli@diagnosticabeniculturali.it
2   Istituto CNR-IRET, Via G. Marconi 2, 05010 Porano, Italy
3   Dipartimento di Chimica, Biologia e Biotecnologie, Università Degli Studi di Perugia, Via Elce di Sotto 8, 06123 Perugia, Italy; sgamellotti.antonio@gmail.com
4   Accademia Nazionale dei Lincei, Via della Lungara 10/230, 00165 Rome, Italy
*   Correspondence: m.vagnini@diagnosticabeniculturali.it (M.V.); chiara.anselmi@cnr.it (C.A.)

**Abstract:** Non-invasive optical spectroscopical analyses were conducted on the three main walls of Alexander and Roxane's Wedding Room in Villa Farnesina, Rome. The north and the east walls were frescoed by Sodoma in 1519. The decoration of the third wall was subsequent and neither the author nor the period is known. The north and east walls underwent various restorations, some even very invasive. For these reasons, the supposed remaining original parts of the two walls by Sodoma were studied and compared with the third one, aiming to obtain more information about its author and epoch. The results show the use of the same pigments for the three walls. In particular, the same yellow pigments including lead antimonate, the use of enamel blue with Bi impurities whose use is time-limited, and the use of a certain kind of purple hematite. The commonality in the pictorial technique also emerged, especially in the yellow parts, painted in the same way on each wall. This information, and documentary sources, reinforce the hypothesis that the third wall was decorated shortly after the death of Agostino Chigi by someone who was well-acquainted with the materials and techniques used by Sodoma for the other two walls.

**Keywords:** non-invasive investigations; portable/reflectance spectroscopy; portable Raman spectroscopy; lead antimonate; enamel blue; caput mortuum

## 1. Introduction

Villa Farnesina, the Renaissance Roman villa built in 1506 by the Sienese banker Agostino Chigi, patron and close friend of Raphael, is currently the headquarters of the Accademia Nazionale dei Lincei, which is devoted to promoting and protecting its huge artistic heritage consisting in some of the most famous artworks of the Italian Renaissance Masters—Raphael included—which were summoned by Chigi for adorning his own private, suburban, relaxing place. Among the artists who left their sempiternal mark inside the residence was Giovanni Antonio Bazzi, better known as Sodoma (1477–1549), who frescoed the bridal suite of Agostino Chigi and his beloved Francesca Ordeaschi. The suite was decorated with scenes from the life of Alexander the Great, including his marriage with Roxane, in explicit reference to that of Chigi. Before the new restoration of the room began, some non-invasive analyses were carried out on the two walls painted by Sodoma and his workshop depicting the *Darius family before Alexander* on the east wall (Figure 1) and *Alexander meeting his new bride-to-be Roxane* on the north wall (Figure 2). A third wall, the west one, depicts the *Taming of Bucephalus* by an unknown author (Figure 3). It was frescoed later than the other two walls because it hosted the double bed of Agostino Chigi and Francesca Ordeaschi. After the death of both of them, in 1520, the wall was decorated, but it is still uncertain how much later this work was carried out and by whose hand. Taking advantage of an imminent restoration campaign involving the northern and eastern

walls, the non-invasive analyses were also extended to the *Taming of Bucephalus,* the west wall, which was never investigated before, aiming to collect information about its painting material to be compared with the other two walls of certain authorship in order to establish the epoch of this fresco. It is important to highlight that both the north and the east wall underwent many restorations, three of them documented. The first restoration dates back to 1870, when the Duke of Ripalta, shortly after buying the Villa in emphyteusis [1], had these murals restored, adopting as a programmatic choice to "entrust the restoration of the frescoes by Sodoma, not to an illustrious artist—who no doubt would have hardly resisted the desire to redo as much as possible but, on the contrary, to a modest painter—already elderly—whose action would be limited to the indispensable" [2]. The second one, carried out at the beginning of the 20th century, is documented by an inscription on one of the walls of the room: "Vito Mameli restored in May 1915" whose work remained a "thick dark mixture painted with glue", which the Istituto Centrale del Restauro (ICR) was called to remedy in the 1970s, as can be seen from the technical report attached to the cost report no. 4 of 1 February 1974 signed by the restorer Paolo Mora, which provided for "consolidation, cleaning and restoration of the frescoes by Sodoma and other 16th century artists" and referred to the "implementation of the work plan prepared for the restoration of the entire pictorial decoration of the Villa" [3].

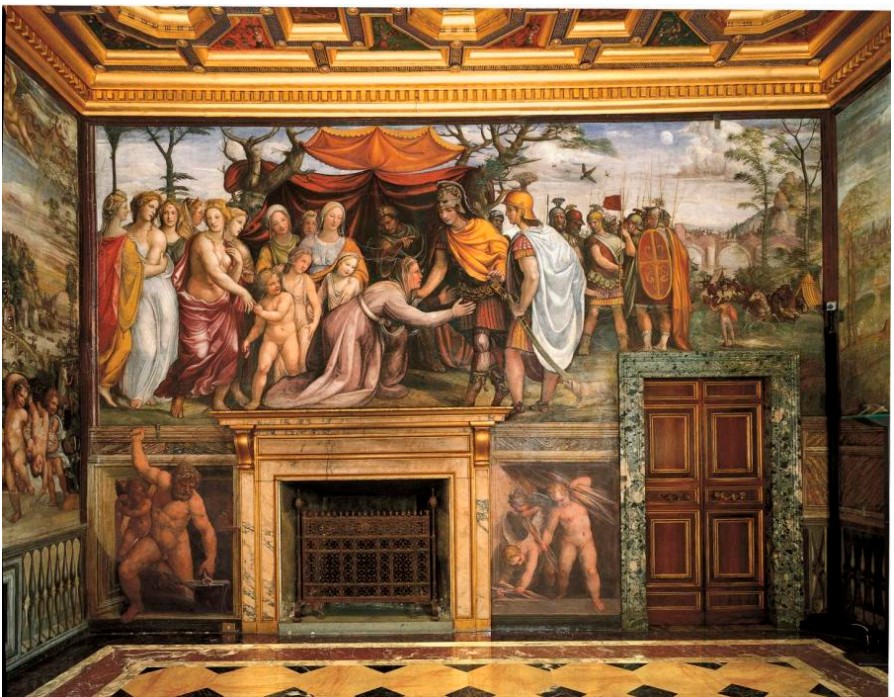

**Figure 1.** The *Darius family before Alexander* Fresco by Giovanni Antonio Bazzi, known as Sodoma, 1519. Alexander and Roxane's wedding room, north wall. Villa Farnesina, Rome. Courtesy of Archivio Villa Farnesina.

For this reason, the non-invasive investigations of the east wall depicting the *Darius family before Alexander* and on the north one depicting *Alexander meeting his new bride-to-be Roxane* were limited to the supposed original parts and were compared with the results emerged from the third wall, the west one, showing the *Taming of Bucephalus* by an unknown artist, trying to chronologically locate such fresco, in order to establish whether it was coeval to the other two.

A set of non-invasive optical spectroscopic analyses, including Visible Reflectance (Vis-R) portable Raman and Reflectance Infrared Spectroscopy (MIR), have been used in a multi-technique approach, widely used and well-established in the last decade [4,5] for the individuation of painting materials by refining and unravelling the information

about chemical elements from X-Ray Fluorescence (XRF), making possible the comparison among the pigments of the three walls and even disclosing some peculiarities of the execution technique.

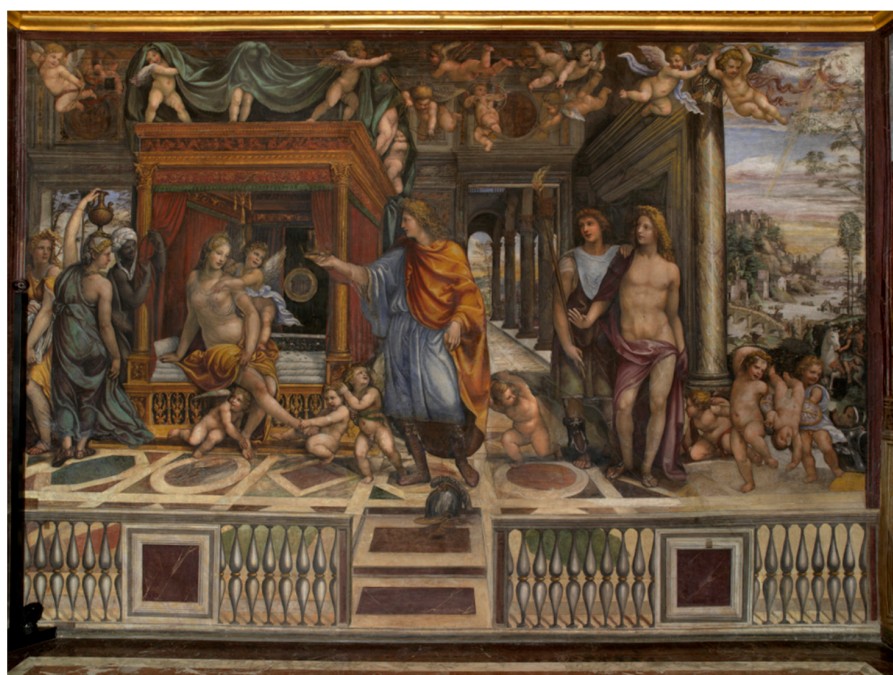

**Figure 2.** *Alexander meeting his new bride-to-be Roxane*. Fresco by Giovanni Antonio Bazzi, known as Sodoma, 1519. Alexander and Roxane's wedding room, east wall. Villa Farnesina, Rome. Courtesy of Archivio Villa Farnesina.

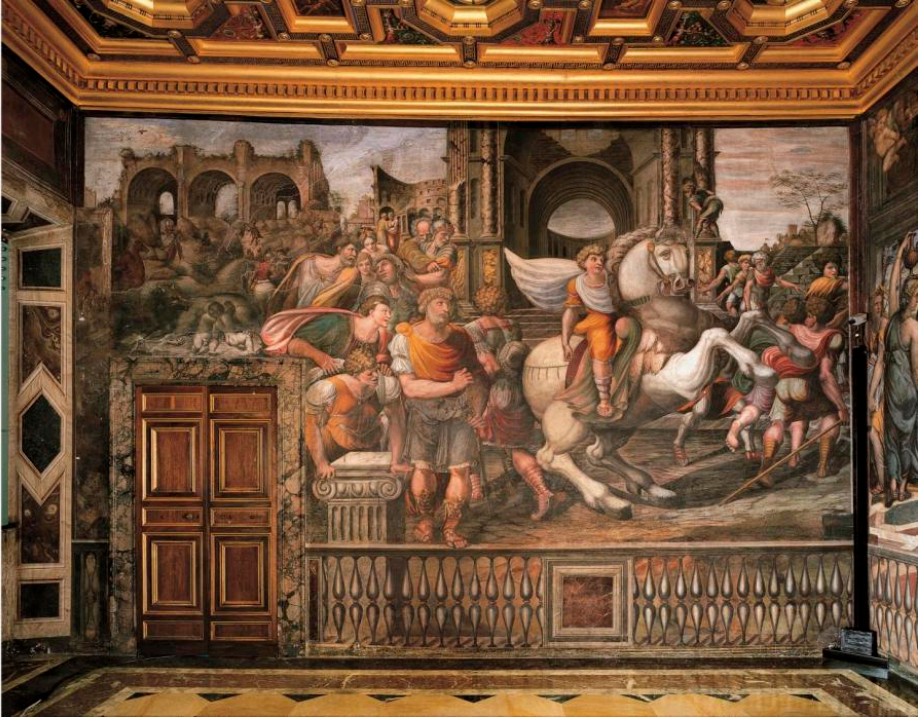

**Figure 3.** *Taming of Bucephalus*. Unknown author and date. Alexander and Roxane's wedding room, west wall. Villa Farnesina, Rome. Courtesy of Archivio Villa Farnesina.

## 2. Materials and Methods

### 2.1. X-ray Fluorescence Spectrometer (XRF)

The portable XRF instrument Tracer III-SD (Bruker AXS) consists of an X-ray tube equipped with a Rh target, and a Peltier cooled Si drift XFlash detector having a resolution of 130 eV FWHM at 5.9 keV. The source was operated at 40 kV and 0.030 mA, with a data acquisition time of 30 s. This instrumental setup allows for the analysis of elements with an atomic number (Z) greater than 10. The X-rays emitted by the tube are collimated on the analyzed surface with a spot diameter of 4 mm. The spectra, corrected for the efficiency of the detector, were expressed as counts per second (cps).

### 2.2. Portable Raman Spectrometer

The BRAVO spectrometer uses a new patented technology called SSE™ (Sequentially Shifted Excitation, patent number US8570507B1) in order to mitigate fluorescence [6,7]. The laser is slightly wavelength-shifted during the acquisition three times, and three raw Raman spectra are recorded. A proper algorithm recognizes all the peaks that shift at different laser wavelengths as good Raman peaks, and other peaks, non-shifting, as fluorescence (or absorbance) peaks, removing them. Moreover, the BRAVO use two different lasers (DuoLaser™), ranging from 700 to 1100 nm, during the acquisition. The use of the second laser is not intended as in usual commercial handheld or portable Raman spectrometer as a tool to try to mitigate the fluorescence, but as a way to collect Raman spectra up to $3200 \text{ cm}^{-1}$ and hence to access the CH stretching region also. The first laser is dedicated to the acquisition of the Raman spectra in the first range (called fingerprint region), and the second one in the second range (called CH region). The BRAVO acquired spectrum is finally a Raman spectrum free from fluorescence on a whole spectral range from 3200 to $300 \text{ cm}^{-1}$. The applied laser power is always less than 100 mW for both lasers. Obviously, using the two BRAVO lasers ranging from 700 to 1100 nm, the sensitivity to inorganic green and blue is very reduced compared to the use of a 532 nm laser. The spectral information from the enhanced spectral range are useful for identification, for example, of resins and waxes. The spectra were acquired with acquisition time ranging from 500 ms to 2 s and accumulation ranging from 5 to 100. For all the measurements, OPUS™ software (Version 7.7) has been used in order to select the appropriate acquisition parameters. We performed at least 4 acquisitions for each point.

### 2.3. Visible Reflectance

The visible reflectance measurements have been carried out by a portable spectrometer CM-700d produced by Konika Minolta. The spectrometer is equipped with an UV radiation filter Xenon lamp and a silicon photodiodes array detector. The analysis range is 400–700 nm with a slit of 10 nm. Illumination area: $8 \text{ mm}^2$, Observer: 10, Illuminant: D65, measurement conditions: SCI (SCI/SCE, illumination/acquisition geometry), number of averaged acquisitions: 5. We performed at least 4 acquisitions for each point.

### 2.4. Reflection Infrared Spectroscopy

The portable infrared spectrophotometer ALPHA-R (Bruker Optik GmbH) is equipped with a Globar IR source, a patented interferometer (RockSolid™, insensitive to external vibrations and able to work in any spatial orientation), and a DLaTGS room temperature detector. The working optical layout for reflection measurements is 22°/22° (specular optics), with about 15 mm of working distance. The infrared spectra have been acquired in the spectral range $7500–375 \text{ cm}^{-1}$ with a spectral resolution of $4 \text{ cm}^{-1}$ and 200 interferograms. The sampling area was $28 \text{ mm}^2$. A background correction using a reference spectrum from a gold flat mirror was applied for representing the reflectance profile (R), expressed in the graphs as pseudo-absorbance, log (1/R).

## 3. Results

Non-invasive analyses were carried out on the supposed unrestored parts of the north and east walls of Alexander and Roxane's wedding room in Villa Farnesina, both attributed to the painter Giovanni Antonio Bazzi, called Sodoma, and his workshop. Further analyses were also carried out on the third wall, the west one, of an unknown period and author in order to compare the results. The Tables 1–3 show the colors investigated and the main results for each employed technique. Not every color of the frescoes has been analyzed because a large part of the walls underwent heavy restorations, whose traces are also still sometimes visible in the investigated areas. Therefore, we limited our investigation to those supposed unrestored parts whose colors are shared by each of the three walls. The XRF analysis indicates Fe and Co as the most abundant elements. Fe, which often characterizes most of the pigments used for the fresco technique because of their stability and coverage, is found in red, orange, yellow, and purple areas. Cobalt is present instead as the main constituent of blue areas, widely used in all the walls studied both alone and in a mixture with other pigments.

### 3.1. Red

In the red areas, the presence of Fe-based pigments prevails (Table 1). In the darker shades, Raman spectroscopy identified red ochre, with the addition of a few hematite (Figure 4a, right, line 2). The bright red of Dario's daughter's dress at the left edge of the east wall is characterized by Fe (XRF, Figure 4a, left, line 1), but Raman spectroscopy individuates only calcium carbonate with its band at 1085 cm$^{-1}$ due to the symmetric stretching of C-O bonds [8], while hematite results instead in the dark red areas, such as the skirt of Alexander (Figure 4a, line 2). In Figure 4a, the characteristic Raman bands of hematite at 498 and 610 cm$^{-1}$ are highlighted [9].

A similar mixture of generic Fe-based red with hematite can be found in the north wall on the red curtains of Roxane's canopy bed on the north wall and in the dress of the putto on the right edge (Table 2). The first derivative of the reflectance spectrum of the red curtains (Figure 4b, line 3) does not show differences with respect to the bright red ones of the east wall (Figure 4b, lines 1, 2, areas 1, 2), and in each analyzed point, the spectrum follows the trend of a generic iron oxide. It is worth noting that on the curtains (Figure 4b, area 3), traces of cinnabar were also found, confirmed by Raman spectroscopy [17] and by Hg from XRF. Since from the reflectance spectrum it is not possible to individuate the features of cinnabar, it is likely that it was used in a mixture and is not found enough on the surface so as to be revealed. The third wall shows again the use of Fe-based red and hematite in the skirt of the back-turned man running away behind Bucephalus (Table 3). The reflectance spectrum of this area (Figure 4b, line 4/area 4) is similar to those recorded on the other two walls.

**Table 1.** Main results of non-invasive investigation on the east wall—*The Darius family before Alexander* (Ip: Inflection point, Max: maximum, Min: minimum).

| Wall | Colored Area | X-ray Fluorescence (XRF) Main Elements | Vis-Reflectance (Vis-R) | Raman | Medium Infrared (MIR) | Comments |
|---|---|---|---|---|---|---|
| *The Darius family before Alexander* (east wall) | Red | Ca, Fe, Sr, K (bright red) | Ip:580,640 nm | $CaCO_3$ [8] | | Generic Fe-based red from Vis-R [9] |
| | | Fe, Ca, K, Sr (dark red) | Ip:580,640 nm | Hematite [10] | | Generic Fe-based red from Vis-R [9] |
| | Yellow | Ca, Fe, Pb, Sr, Sb (bright yellow) | Ip:440,540,630 nm | Yellow ochre [11] | | From Vis-R yellow ochre [12] + Naples yellow [12] |
| | | Ca, Fe, Pb, Sr (dark yellow) | Ip:451,533,640 nm | | | Mostly Pb-yellow [12] +yellow ochre from Vis-R |
| | Purple | Ca, Fe, K, Sr | Ip:589,640,695 nm | $CaCO_3$ [8], hematite [10] | $CaCO_3$ [8], silicates [13] and acrylic resin [14] | Hematite-based purple, possible caput mortuum [15] |
| | Purple/greyish | Ca, Fe, Co, K, Si, As, Bi, Sr | Ip:589,640,690 nm | $CaCO_3$ [8], hematite [10] | | Hematite-based purple, possible caput mortuum [15]. Smalt is not visible in Vis-R |
| | Blue | Ca, Co, Fe, Pb, As, K, Bi, Sr, Si | Max:468,561,622 nm; Min:530,590,650 nm; Ip:687 nm | $CaCO_3$ [8] | | Smalt [16] |
| | | Ca, Co, Fe, Pb, As, Ti, K, Bi, Si | Max:468,561,622 nm; Min:530,590,650 nm; Ip:687 nm | | | Smalt [16] and traces of restoration materials (Ti) |

**Table 2.** Main results of non-invasive investigation on the north wall—*Alexander meeting his new bride-to-be Roxane* (Ip: Inflection point, Max: maximum, Min: minimum).

| Wall | Colored Area | X-ray Fluorescence (XRF) Main Elements | Vis-Reflectance (Vis-R) | Raman | Medium Infrared (MIR) | Comments |
|---|---|---|---|---|---|---|
| *Alexander meeting his new bride-to-be Roxane* (north wall) | Red | Fe, Ca, K, S, Sr, Pb, Hg | Ip:580, 640 nm | Cinnabar [17] | | Generic Fe-based red from Vis-R [9] Cinnabar from XRF and Raman, but not visible by Vis-R. |
| | | Ca, Fe, Sr, K, S, Pb | Ip:583, 640 nm | | | Generic Fe-based red from Vis-R [9] |
| | Orange | Fe, Ca, K, Sr, Pb | Ip:434,510,579,636 nm | CaCO$_3$ [8] + yellow ochre [11] | . | Mixture of red and yellow ochre. Yellow ochre visible from Raman and Vis-R. Pb-yellow not to be excluded due to the inflection at 510 nm from Vis-R [12] |
| | Yellow | Ca, Fe, K, Sr, Pb Fe, Ca, Pb, K, Sr, Sb Ca, Pb, Fe | Ip: 440, 544, 640 nm Ip: 436,541,631 nm Ip:450, 519,631 nm | Pb-antimonate [18,19] | | Only yellow ochre from Vis-R Yellow ochre+Naples yellow from Vis-R Mostly Pb-yellow from Vis-R [12] |
| | | Pb, Sb, Fe (K) | Ip:443,508,546,640 nm | Pb-antimonate [18,19] | | Naples yellow+Pb-yellow +yellow ochre from Vis-R [12] |
| | Purple | Fe, Ca, K, Sr, S, | Ip:589,640,695 nm | Hematite [10] | | Hematite-based purple, possible caput mortuum [15] |
| | Blue | Ca, Fe, K, Sr, S | Max:477 nm, Min:603 nm, Ip:683 nm | CaCO$_3$ [8] | CaCO$_3$ [8], weak signal of lapislazuli [20], acrylic resin [14] | Lapislazuli from Vis-R [12] and traces of restoration materials |
| | | Ca, Fe, Co, As, K, Si, Sr, Bi | Max:480 nm, Min:595 nm, Ip:680 nm | | | Lapislazuli and smalt [16] in different superimposed layers. From Vis-R only Lapislazuli [12] |
| | | Ca, Fe, Co, As, K, Sr, Bi | Max:481 nm, Min:541,590,637 nm, Ip:680 nm | | | Lapislazuli [12] and smalt [16] mixed together; Vis-R spectrumhas both the features of these pigments |
| | | Ca, As, Co, Fe, K, Bi, Sr, Pb, Si | Max:470,574,623 nm, Min:524, 593,652 nm, Ip:685 nm | | | Smalt [16] |
| | | Ca, Fe, As, Co, K, Sr, Bi | Max:478 nm, Min:576 nm, Ip:688 nm | | | Lapislazuli-like spectrum in Vis-R [12]; smalt visible only in first derivative of Vis-R |

**Table 3.** Main results of non-invasive investigation on the west wall—*The Taming of Bucephalus* (Ip: Inflection point, Max: maximum, Min: minimum).

| Wall | Colored Area | X-ray Fluorescence (XRF) Main Elements | Vis-Reflectance (Vis-R) | Raman | Comments |
|---|---|---|---|---|---|
| *The Taming of Bucephalus* (west wall) | Red | Ca, Fe, K Sr | Ip:582,640 nm | CaCO$_3$ [8] | Generic Fe-based red from Vis-R [9] |
| | | Fe, Ca, Sr, K, S (dark red) | Ip:580,640 nm | Hematite [10] | Generic Fe-based red from Vis-R [9] |
| | Orange | Fe, Ca, Sr, Pb, K | Ip:435,510,570,635 nm | CaCO$_3$ [8] | Mixture of red and yellow ochre; Pb-yellow not to be excluded due to the inflection at 510 nm [12] |
| | Yellow | Pb, Fe, Sb (K) | Ip:450,521,627 nm | Pb-antimonate [18,19] | Pb-yellow and few yellow ochre from Vis-R [12]. Naples yellow from XRF and Raman |
| | | Pb, Fe, Sb | Ip:449,506,627 nm | Pb-antimonate [18,19] | Mostly Pb-yellow from Vis-R [12] |
| | Purple | Fe, Ca, Sr, K | Ip:590,639,688 nm | CaCO$_3$ [8], hematite | Hematite-based purple, possible caput mortuum [15] |
| | Blue | Ca, Co, Fe, Bi, As, K, Si, Pb | Max:440,549,608 nm; Min:500,580,637 nm; Ip:692 nm | CaCO$_3$ [8] | Smalt [16] |

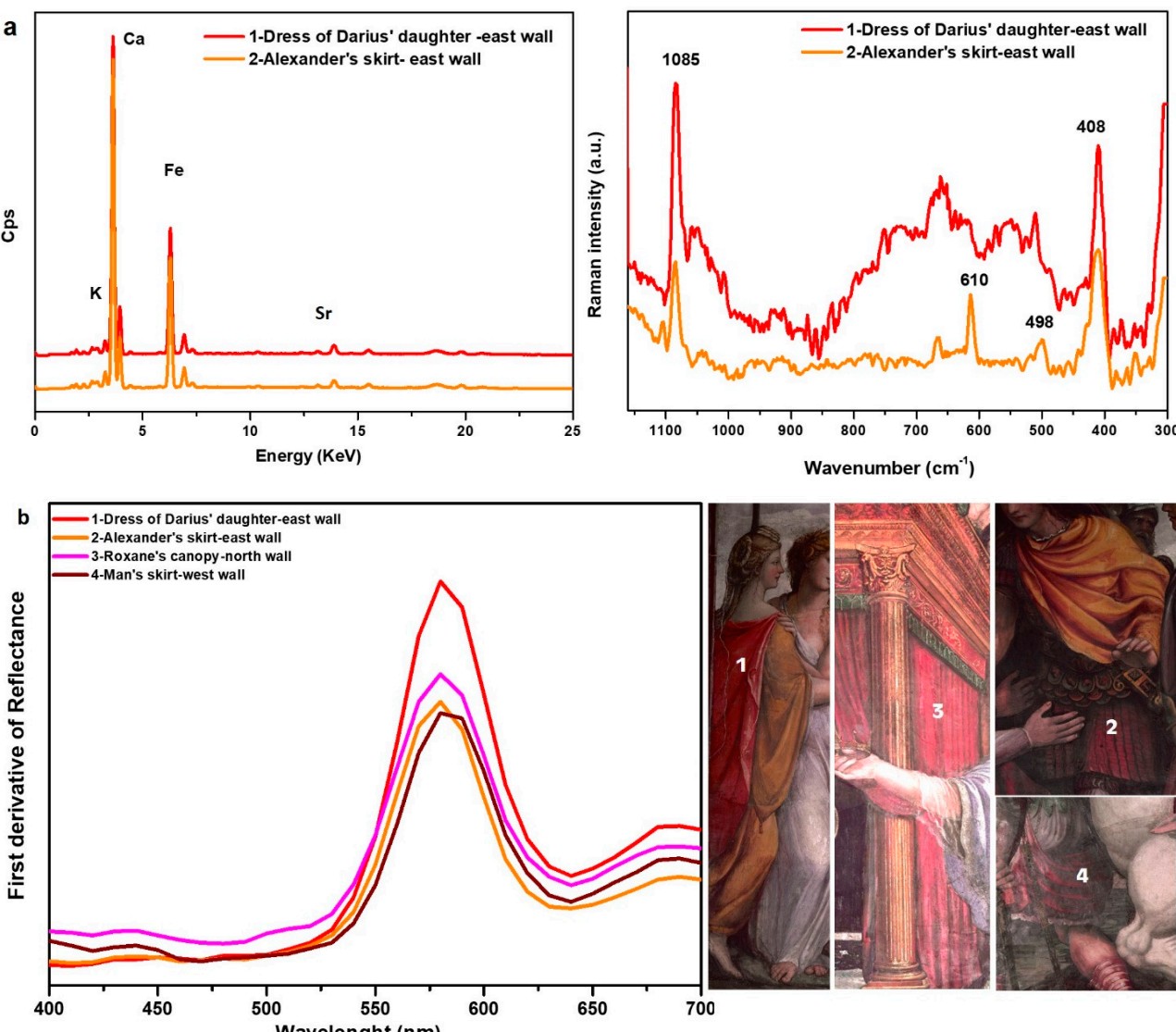

**Figure 4.** (**a**) From left to right: XRF and Raman spectra of points 1 and 2 on the east wall. (**b**) Comparison of the first derivative reflectance spectra measured on selected red areas (lines 1–4).

### 3.2. Purple

The composition of the purple areas by XRF is the same as the red ones on each of the three walls (see Tables 1–3). Raman spectroscopy instead individuates stronger hematite signals with respect to the red ones in all the purple zones, such as in the dress of Dario's wife on the east wall (Table 1).

The dress of Darius' wife Statira II appears as red/purplish (Figure 5, area 2) and has the same elemental composition as the red dress of Darius' daughter (Figure 5, area 1) depicted next to her (Table 1, bright red), but Raman spectroscopy indicates a strong presence of hematite only in the former (Table 1, dark red).

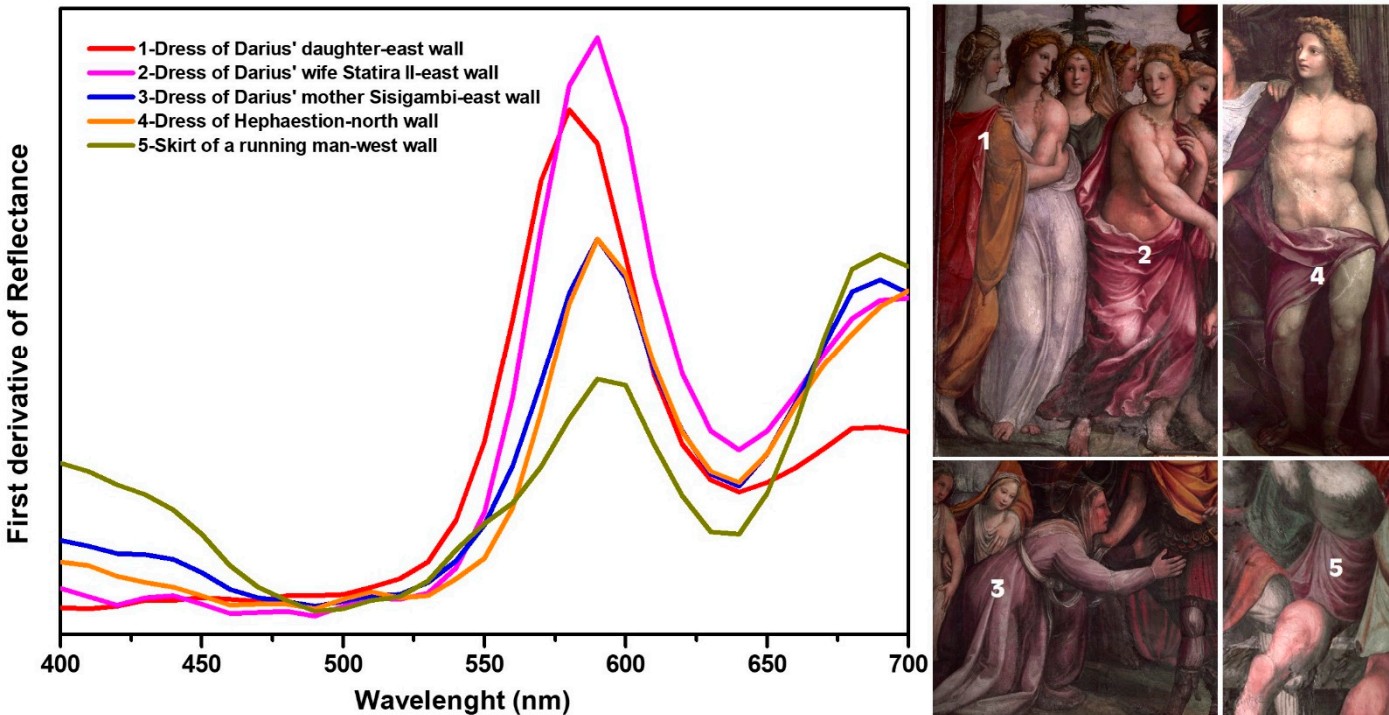

**Figure 5.** Comparison of first derivative reflectance spectra measured on selected red and purple areas (lines 1–5).

This difference of hue is hardly appreciated in the reflectance spectra but becomes clearer when considering their first derivatives. Indeed, for the dress of Darius' wife Statira II (Figure 5, line 2), there is a shift of the maximum of about 10 nm towards the red, and at the same time, the narrowing of the minimum to 640 nm and the consequent steep climb up to 700 nm with respect to the daughter's red dress (Figure 5, line 1). These features recur in the purple areas of each of the three walls, and more particularly, in the already discussed robe of Statira II on the east wall (Figure 5, line 2), of the half-naked Hephaestion on the north wall (Figure 5, line 4/area 4), and on the skirt of a running, back-turned figure on the third wall, the Taming of Bucephalus (Figure 5, line 5/area 5).

The purple/greyish dress of Darius' mother Sisigambi (Figure 5, area 3) consists of hematite and smalt according to a common technique concerning frescoes [15].

Smalt is indicated by XRF from the Co presence and its associated impurities, such as Bi and As. The presence of smalt is not identifiable in the reflectance spectrum, which follows exactly those of Co-free, hematite-based purples, so this area was probably painted with different superimposed layers, with purple hematite on the surface.

### 3.3. Yellow

The yellow areas are quite articulated, being a mixture of two or three different yellow pigments. Their composition is mostly of two types: one containing Fe and Pb, and the other Fe, Pb, and Sb. The simultaneous presence of Sb and Pb indicates the use of Pb-antimonate, also known as Naples yellow [21,22], and always coincides with lighter yellow hues. On the east wall, the use of Naples yellow is found on the lighter parts of the dress of Darius' daughter (Figure 6, area 1), where XRF indicates Fe, Pb, and Sb (Table 1). The first derivative of its reflectance spectrum is mostly that of yellow ochre (Figure 6, line 1), but a little broader, according to Naples yellow features [12]. In the yellow skirt of Hephaestion, XRF indicates Fe and Pb but no Sb. Since there is no evidence of lead white in this area, Pb is probably due to litharge, a lead monoxide, as can be inferred in the first derivative of the reflectance spectrum from the maximum at 530 nm [12], mixed with yellow ochre, which can be seen in its weakened maximum at 450 nm (Figure 6, line 2). An abundant use of litharge still accompanied by a background of yellow ochre, as suggested by Fe

presence, is even clearer in the landscape on the north wall (Figure 6, area 4), where XRF indicates only three main elements: Ca, Pb, and Fe (Table 2), and the reflectance shows, in its first derivative, mostly the spectrum of litharge but little modified by yellow ochre, recognizable for the maximum at 450 nm (Figure 6, line 4). The yellow dress of Roxane's maid on the left edge is characterized by Fe and Pb, and the first derivative of its reflectance spectrum essentially shows yellow ochre, so in this case, the source of Pb should be other than litharge (Figure 6, line 3/area 3). All three types of pigments, yellow ochre, litharge, and Naples yellow, can be found in Roxane's dress (Figure 6, area 5), where Naples yellow is used to paint lighter, cold tones. The main elements found from XRF, Fe, Pb, and Sb, suggest the presence of Naples yellow and yellow ochre, but from Vis-reflectance, litharge can also be individuated.

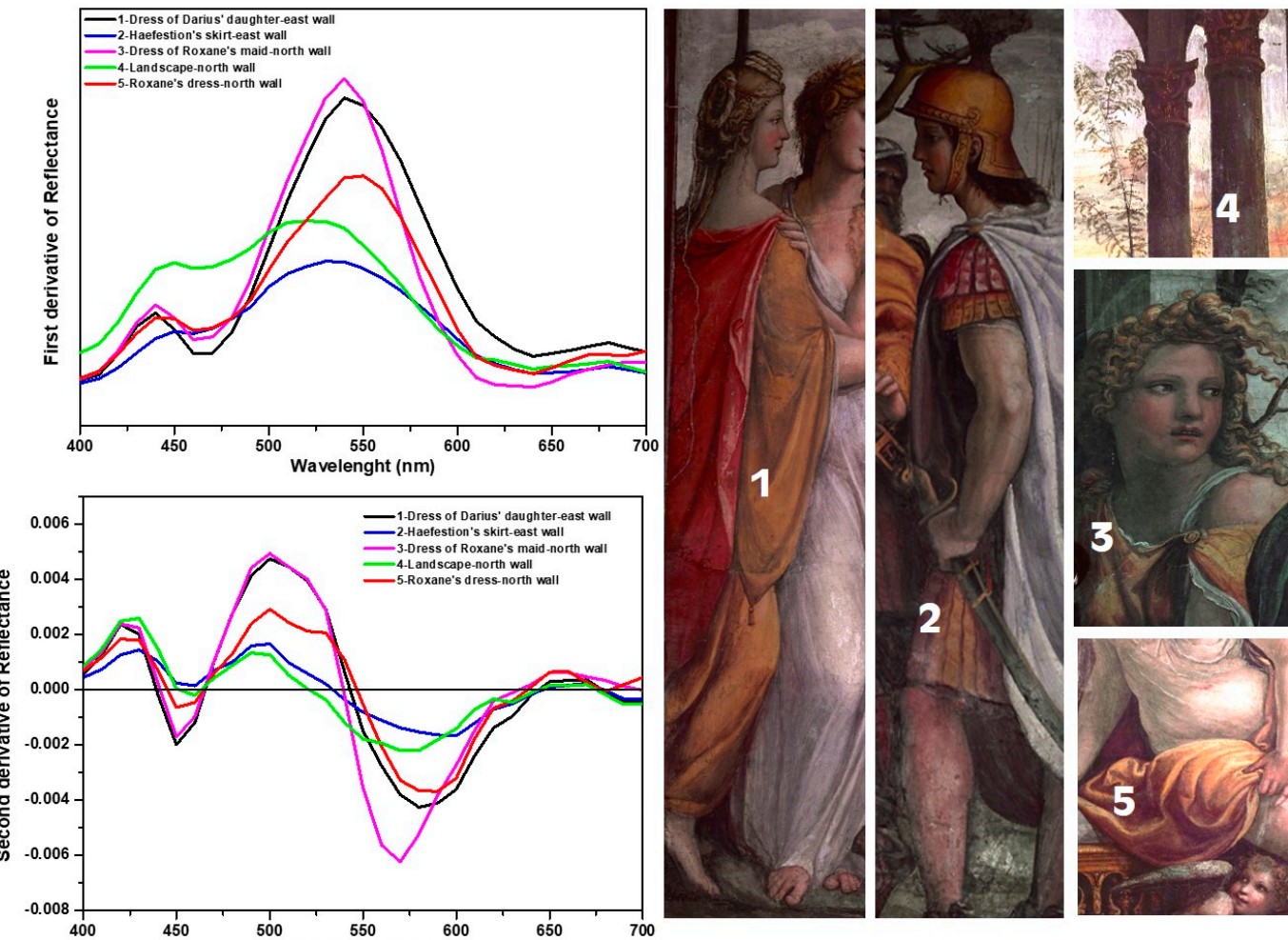

**Figure 6.** Comparison of first (**top**) and second (**bottom**) derivative reflectance spectra, recorded on selected yellow areas (lines 1–5).

Litharge's presence can be argued by the first derivative (Figure 6, line 5), where Roxane's dress shows the main features of both yellow ochre and Naples yellow: namely, the relative maximum at 445 nm and the broader shape of the main band, respectively. Nevertheless, a further growth at 509 nm (Figure 6, line 5) suggests the presence of litharge, otherwise impossible to establish on the basis of XRF data alone, due to the joint presence of Pb and Sb. Litharge features of Roxane's dress are more evident if the second derivatives are checked, in the maximum at 494 nm when compared with the landscape area, which was the one with the most evident presence of litharge (Figure 6, lines 4, 5). In fact, if we consider the second derivatives, the mixture of yellow ochre and Naples yellow (Figure 6,

line 1) does not account for the trend between 465 and 550 nm of Roxane's dress (Figure 6, line 5), which can be explained by the presence of litharge already seen in the landscape (Figure 6, line 4) with the two relative maximums at 494 and 529 nm, which also appear in this spectrum, facilitating the identification of litharge in a situation difficult to decipher due to the overlapping of common elements. Naples yellow is also easily individuated by its Raman features at 510 and 655 cm$^{-1}$ [11,12], together with calcium carbonate [8] (Figure 7). The 655 cm$^{-1}$ Raman band of Naples yellow is present when there is an excess of lead and it is specific to a non-stoichiometric Pb:Sb molar ratio [11].

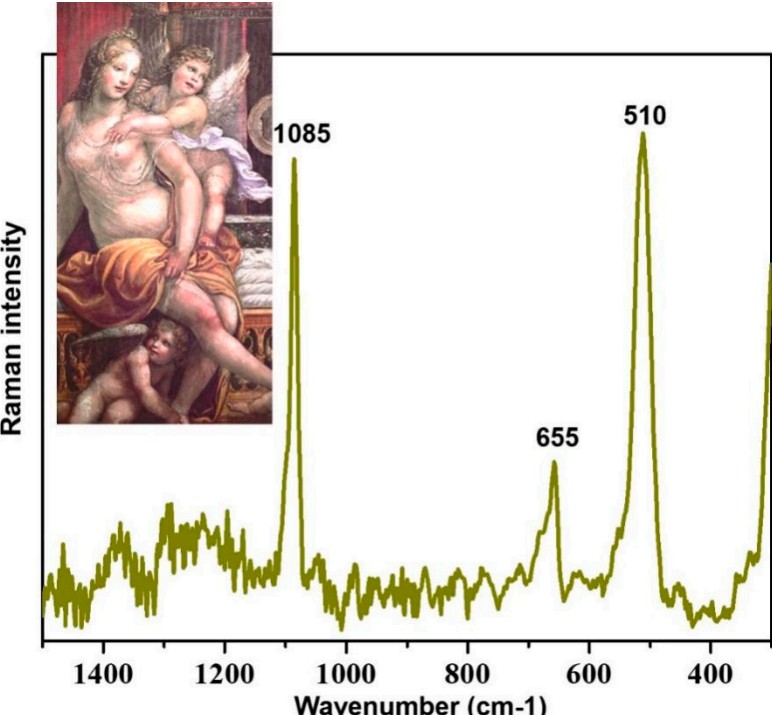

**Figure 7.** Raman spectrum of Roxane's yellow dress showing the presence of lead antimonate and calcium carbonate.

The presence of Naples yellow is also found on the third wall, as revealed by XRF and Raman spectroscopy (Table 3). However, in these areas, the first derivative of reflectance spectra almost exclusively shows the shape of litharge (Figure 8, lines 3, 4), as can be seen from Figure 8 by comparison with the reflectance spectrum on the landscape of the north wall (mostly litharge and yellow ochre, Figure 8, line 2) and with the dress of Roxane's maid, which in reflectance shows mostly yellow ochre (Figure 8, line 1). The striking presence of litharge allows for its otherwise difficult identification, due to the simultaneous presence of Pb and Sb.

*3.4. Orange*

In orange areas, the most evident and common feature of the three walls is the disappearance of Sb from the XRF analysis with respect to the corresponding yellow areas (Tables 2 and 3). In Alexander's mantle on the north wall, the use of red ochre added to the yellow ochre was identified. In addition, the presence of Pb was seen by XRF, and the first derivative of reflectance with the hinted maximum at 510 nm (Figure 9, line 1) also suggests, if compared with other litharge-based areas (Figure 9, lines 2, 4), its presence, which is not so strange due to its yellow/orange hue (Figure 9, area 1).

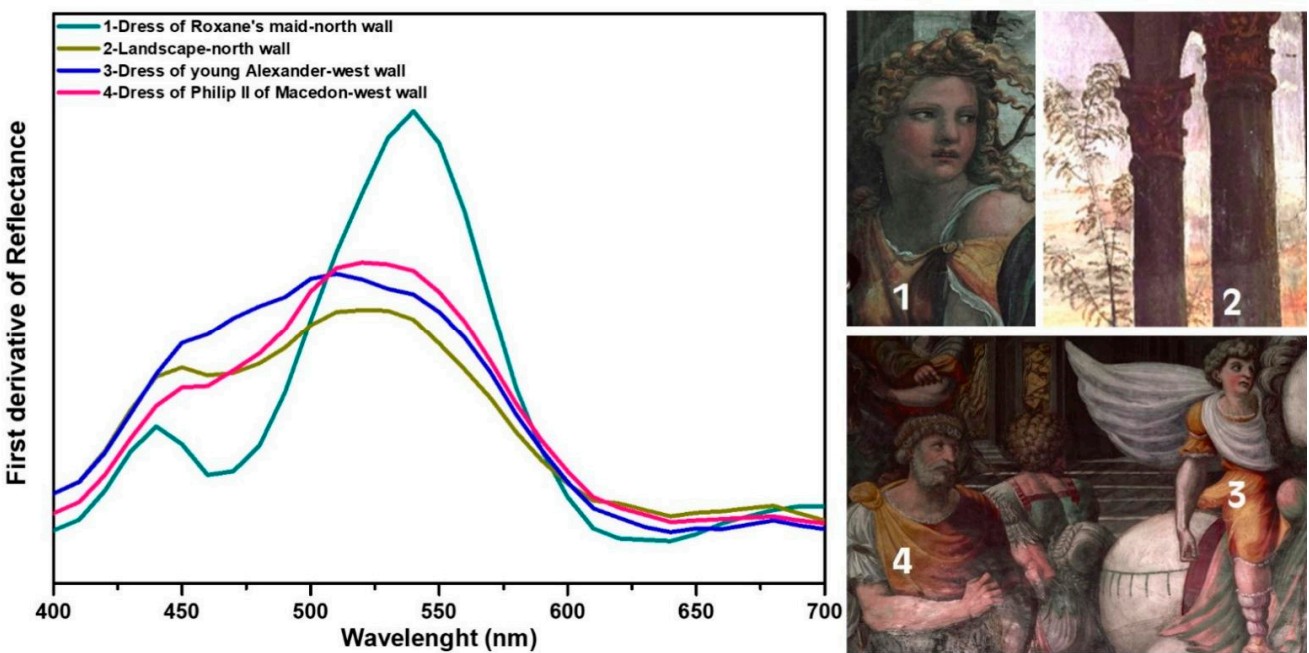

**Figure 8.** Comparison of first derivative reflectance spectra, recorded on selected yellow areas (lines 1–4).

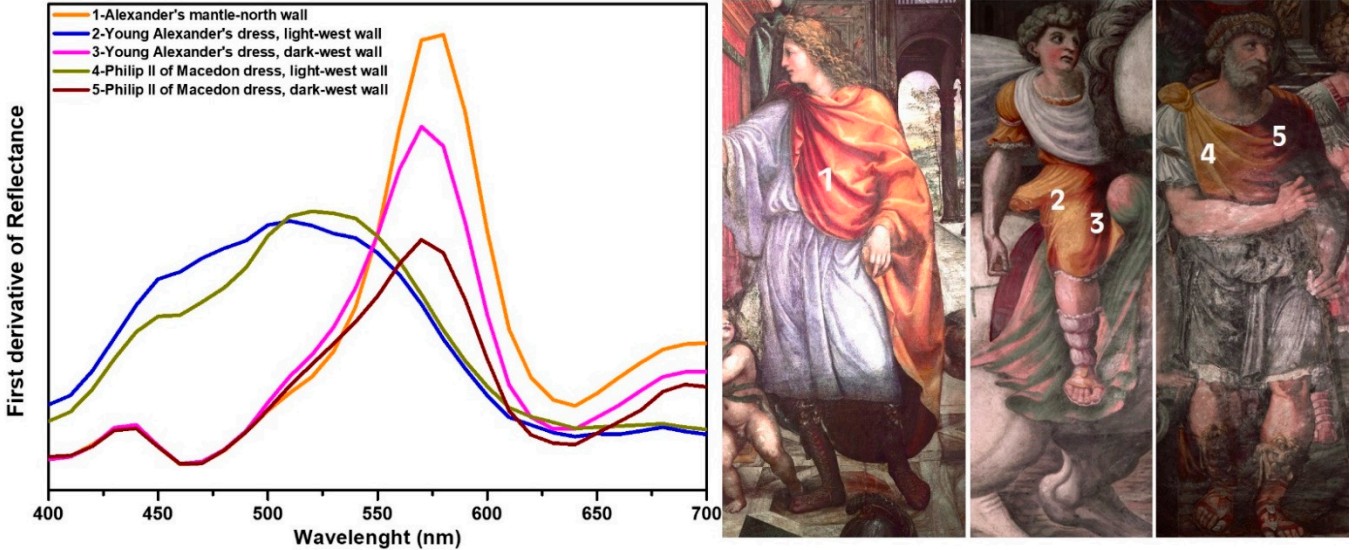

**Figure 9.** Comparison of first derivative reflectance spectra, recorded on selected orange and yellow areas (lines 1–5).

Additionally, on the third wall, the orange areas are characterized by mixtures of yellow and red ochre, with the presence of Pb seen from XRF (Table 3). The use of litharge is also perceivable in the orange hue of the dress of both young Alexander and his father Philip II of Macedon, when considering in the first derivative of their reflectance the weakened maximum at 510 nm compared to their yellow counterparts (Figure 9, lines 2, 3 and lines 4, 5).

### 3.5. Blue

The blue areas in each of the three walls studied are characterized by the presence of Co, which corresponds—in most cases—to the reflectance spectrum of smalt. Along with Co, associated impurities such as As and Bi are always found.

On the east wall, smalt is used to create the shades in the white robe of one of Darius' daughters (Figure 10, area 1) and in the white cloak of Hephaestion (Figure 10, area 2). It

is also found mixed with hematite to obtain the gray/purple hue of Sisigambi's dress, as already highlighted above (Figure 5, area 3). On the third wall, the use of smalt returns, as indicated by the presence of Co, Bi, and As, in the dark blue areas of Alexander's father's robe, Philip II of Macedon (Figure 10, area 4), and in the white garment of the young Alexander taming the horse Bucephalus (Figure 10, area 3). The first derivative of the reflectance spectra measured in the same points confirms its presence (Figure 10).

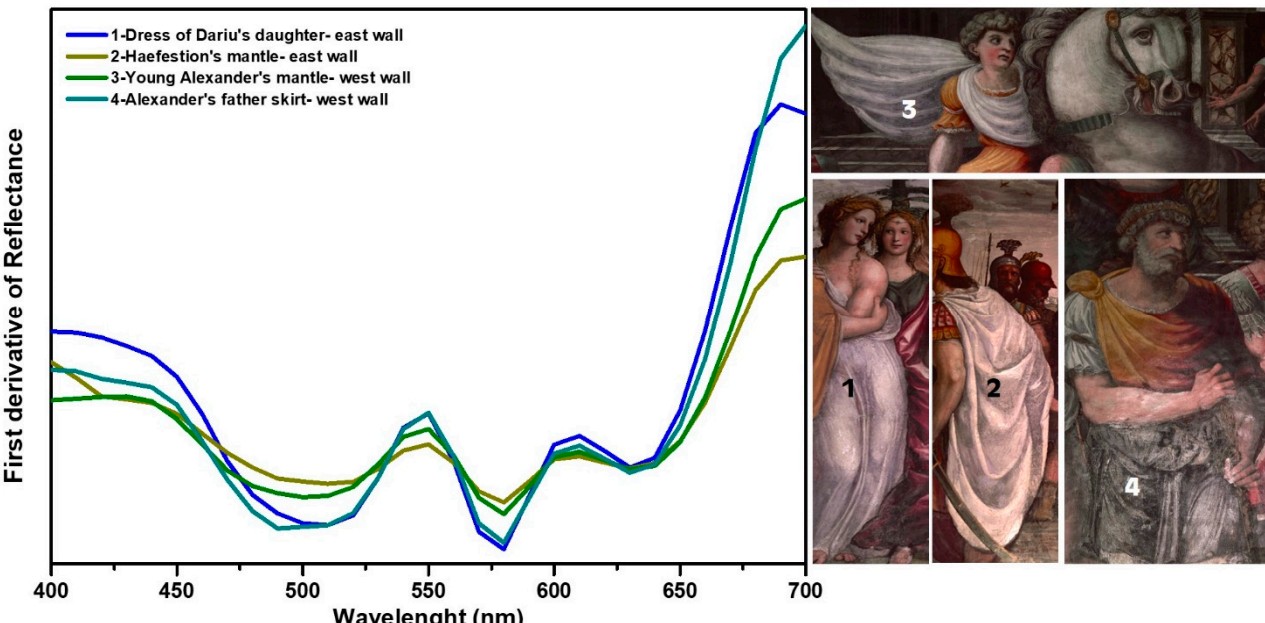

**Figure 10.** Comparison of first derivative reflectance spectra recorded on selected blue areas (lines 1–4).

On the north wall, two different blue pigments were individuated: smalt and lapis lazuli, with the latter univocally identified by infrared spectroscopy (Figure 11) thanks to its characteristic infrared band at 2340 cm$^{-1}$ [20]. Both these pigments have been used to obtain different shades of blue in different ways, namely:

(1) The use of lapis lazuli, only on the north wall, for the blue highlights on white draperies, such as on the white dress of the putto that helps Roxane to get ready (Figure 12, line 1).

(2) The use of smalt. It is found as the only blue for highlighting the white dress of Hymenaeus (Figure 12, area 3): from XRF, the characterizing elements Co, Bi, and As emerged, and the reflectance spectrum corresponds to that of smalt (Figure 12, line 3). It is also found in those areas currently perceived as grey, such as in the helmet in the foreground, on the north wall. XRF indicates the presence of Co, Bi, and As, throughout the whole helmet.

(3) The use of smalt and lapis lazuli in single superimposed layers, with lapis lazuli on the surface, distinguishable only with the joint use of XRF and reflectance spectroscopy, because the latter individuates only pigments on the surface. This is the case of the upper part of Alexander's dress, where the characteristic elements of the blue enamel emerged from the XRF, but its reflectance is almost entirely attributable to that of lapis lazuli, whose elements cannot be revealed by XRF (Figure 12, line 2).

(4) The use of smalt and lapis lazuli mixed together, which is evident from the reflectance spectra of the points analyzed, where the characteristics of both pigments are found. This is the case of the lower part of Alexander's dress and that of the putto on the right, intent on playing with Alexander's shield (Figure 12, line 4).

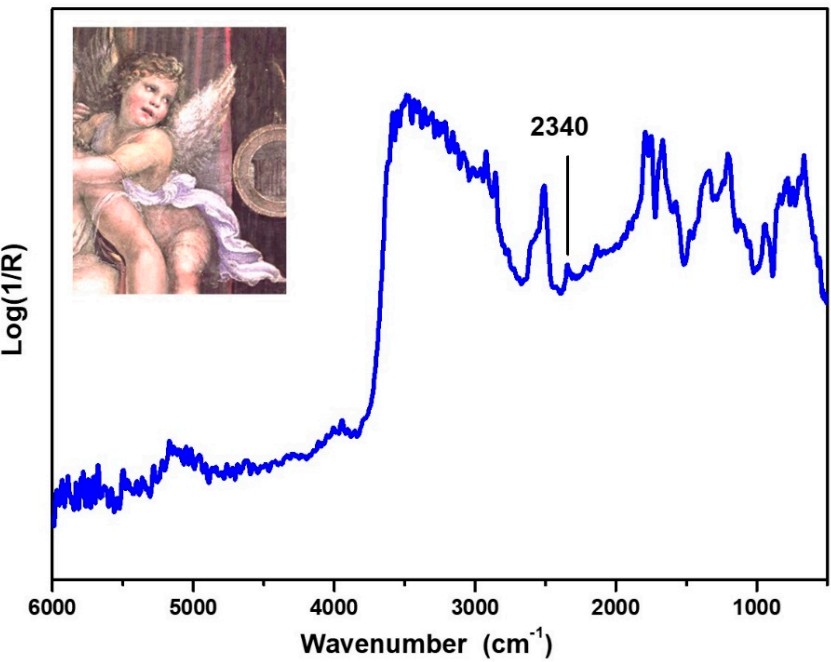

**Figure 11.** Infrared reflectance spectrum recorded on putto's dress, north wall. The lapis lazuli characteristic infrared band is evidenced in the spectrum.

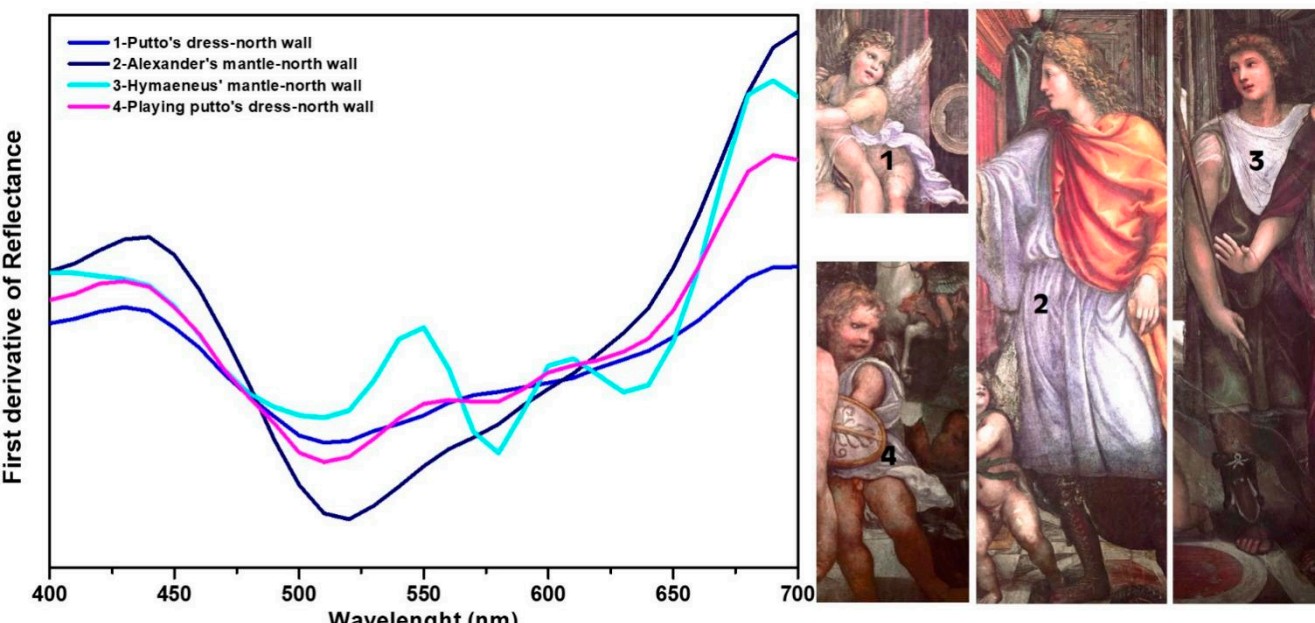

**Figure 12.** Comparison of first derivative spectra of visible reflectance spectra recorded on selected blue areas (lines 1–4).

Further lapis lazuli finishes would also seem by now lost or seriously compromised, as can be seen from the heavy repainting of some mantles, to restore the outlines that are no longer perceptible, including Alexander's dress, which presents elements attributable to restoration interventions such as Ti and Zn. Moreover, some gold finishes which were revealed by XRF only in the north wall also seem to be very compromised by restoration. They are present in the columns, in the mirror behind Roxane, and as labile traces in the sleeves' edge of Alexander's blue garment.

## 4. Discussion

Some common features emerged from the non-invasive analyses of the three walls, which are worthy of discussion. The presence of hematite, identified by Raman spectroscopy, seems to always coincide with a purplish hue. In fact, if we exclude the cases in which hematite is added to a Fe-based pigment to obtain a darker red tone, all the other areas rich in hematite are purple. No organic compound ascribable to a dye has been identified in these purple hues, which show the same composition and the same impurities as the red ones without hematite, and compared to these, show a definitely different shade. This chromatic diversity, with the elemental composition being the same and in the absence of organic dyes, suggests the use of hematite as a purple pigment in its own right, perhaps in the form of caput mortuum [15]. The use of this hematite-based purple pigment occurs in each of the three walls. The yellow areas show a very articulated composition. They are all characterized by the presence of yellow ochre to which other yellow pigments are added. First, Pb-antimonate, also known as Naples yellow, is easy to individuate from Sb and Pb by XRF. Furthermore, from the joint analysis of XRF and visible reflectance, a Pb-based yellow was also revealed, namely litharge, which would otherwise be impossible to identify by the elemental analysis alone when used together with Pb-antimonate. In fact, the contemporary presence of Pb and Sb from XRF may have distracted from considering other Pb-based yellows, but the Vis-reflectance revealed the spectral shape of litharge also in those areas where Pb-antimonate has been found. The presence of the Pb-antimonate or Naples yellow occurs in the yellow areas in the lighter hues, with a cold tone. Naples yellow is one of the synthetically produced pigments known and lost and rediscovered a number of times throughout history [21]. It comes from the glassmakers' tradition and its use as a pigment at the beginning of the XVI century was very early and unexpected [22]. It is worth noting that Naples yellow has already been individuated in Villa Farnesina, in the Loggia of Cupid and Psyche, both in the figurative part of the scenes and in the fruits of the vegetable festoons [23].

In Alexander and Roxane's Wedding Room, it is used in the walls painted by Sodoma to lighten the yellow tones, affording a lighter and cooler shade.

Additionally, on the third wall, the same three yellow pigments were used, but with an interesting difference. On the walls by Sodoma, the Naples yellow is always detectable by reflectance, and this means that it is on the surface, as a "final touch" in the lighter shades. On the third wall, Naples yellow is present in the yellow areas because it is individuated by XRF, but the reflectance does not always detect it, with litharge or yellow ochre prevailing instead.

Thus, even though we found the same yellow pigments for each of the three walls, it is important to highlight the diversity in the "final touch". Indeed, Sodoma used Naples yellow to refine the light hues, while the painter of the third wall, who used the same materials as Sodoma, indifferently mixed Naples yellow with yellow ochre and litharge, without preferring one particular pigment for finishing. Furthermore, the evidence that Sb disappears in the orange areas in both walls one and two, and in the third wall as well, indirectly confirms the precise choice by both Sodoma and the unknown painter to use Naples yellow only for light yellow shades.

Smalt, which is widely used in each of the three walls studied, has the characteristic of having, associated with the Co, some impurities of As and Bi. In particular, the latter allows the pigment to be placed within a specific processing method that obtained the blue of enamel from the bismuth slag, locating its extraction both geographically, as a German manufacture from the Erzgebirge, and chronologically as well, attesting to it within the XVI century [24,25]. This means that the decoration on the third wall is realistically ascribable within the 16th century. It must be remembered that the third wall was the one that housed the double bed of Agostino Chigi and Francesca Ordeaschi, who both died in 1520, *terminus post quem* from which to start the decoration of the third wall. Documentary sources attest the presence of Sodoma in Rome to settle and open a workshop in 1521 [25], and the same sources exclude any kind of pictorial production from Sodoma in the capital from that

date, apart from a few drawings [25]. This fact does not exclude that he may have passed on his knowledge to the students in the workshop, who could have worked inside Villa Farnesina, as their Master did. This information together with the scientific evidence of the painting material support the hypothesis that the third wall is probably coeval to the other two, and painted by someone who had attended Sodoma's workshop and had learned his painting techniques and his painting materials as well.

Finally, it must also be noted that for the north wall, i.e., that depicting Alexander meeting his new bride-to-be Roxane, some precious painting materials such as gold finishes and lapis lazuli were used, which do not appear on the other two walls. These unique features only present on the north wall tell us something about the relevance of this fresco, other than the authorship. In fact, Sodoma used very precious materials for decorating this wall, which he himself did not use for the east one. This must be ascribed to the painted subject of this wall—the marriage of Alexander and Roxane—which must be the most significant of the room and therefore deserving the most precious painting materials which were circulating at that time.

## 5. Conclusions

The results obtained show a substantial uniformity in the typology of the pigments used on the east, north, and west walls of Alexander and Roxane's Wedding Room in Villa Farnesina, Rome. Additionally, the execution technique relative to the mixtures of pigments used was found to be the same, in the rendering of yellows and their different shades from colder tones to golden ones, and in the blue hues on a white background. Although it is not known how much time passed between the execution of the two frescoes, it should be noted that the palette of the east and north walls is the same as the third wall, the west one. In particular, those pigments that in the two walls frescoed by Sodoma are a reason of interest because they were still uncommon or because they were available in a limited period of time, such as Naples yellow, caput mortuum, and smalt, are also found, used in the same way, in the third wall. Combining the spectroscopic evidence resulting from the non-invasive analyses of pigments with historical sources, we hypothesized that the decoration of the third wall, which represents the Taming of Bucephalus, was carried out shortly after 1520 and within the sixteenth century, as evidenced by the use of smalt containing Bi impurities. Moreover, the decoration of the third wall was likely accomplished by an artist who surely knew Sodoma's painting technique very well, as evidenced by the same materials used, and the same execution technique for painting particular areas. This work demonstrates how investigations on pictorial materials can reveal very useful details, much like documentary sources, in order to obtain—totally non-invasively—crucial information about some still uncertain artistic productions.

**Author Contributions:** Conceptualization, C.A. and M.V.; methodology, M.V., M.A. and C.A.; investigation, M.V., M.A., C.A. and A.S.; resources, A.S.; data curation, M.V. and C.A.; writing—original draft preparation, C.A. and M.V.; writing—review and editing, M.V., M.A., C.A. and A.S.; supervision, A.S. All authors have read and agreed to the published version of the manuscript.

**Funding:** This research received no external funding.

**Institutional Review Board Statement:** Not applicable.

**Informed Consent Statement:** Not applicable.

**Acknowledgments:** Virginia Lapenta is gratefully acknowledged for her kind availability, the interesting case which she invited us to study, and the images supplied.

**Conflicts of Interest:** The authors declare no conflict of interest.

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
