# Peer review of "Things Always Come in Three: Non-Invasive Investigations of Alexander and Roxane’s Wedding Room in Villa Farnesina"

_heritage, doi:10.3390/heritage4040157_

Round 1

Reviewer 1 Report

 The manuscript is quite well written and can be published after the following issues have been corrected / improved. The main weakness is the absence of a table listing the identified staining agents / phases, their location and which technique proves the presence of the agent / phase as well as the references used for assignment. A comparison of the use of coloring agents identified in other works of the same period (paintings, frescoes) would greatly increase the interest and impact of the manuscript.

There are also a few minor issues listed below.

Page 2, line 51 spouses (unclear)

Page 5, line 135, -1 superscript, same correction in the X-label of Raman figures.

Page 5:  Raman analysis: what is the size of the analyzed spot?

Line 135 fifirst?? Line 136 flfluorescence ?  Page 6, line 142 identiffication, etc. to be corrected

Note that a baseline correction is performed automatically by the instrument!

Page 6: surface area of the visible reflectance measurement?

It is well known that the depth of penetration of x-rays changes with energy (see for example Heritage, 2020, 3, 1302). The measured depth range should be compared with the thickness of the colored fresco layer for the different elements.

Page 7, line 189 : LO Eu peak ? In the absence of factor analysis of groups, such a mention is unnecessary. In addition, in natural hematite, some Fe ions are usually replaced by Ti or Al ions, which makes the description of the mode more complex.

Fig. 4: mark all the XRF peaks; where is the peak of Hg? The measured Raman wave number range makes it difficult to identify HgS; add some evidence / discussion.

Page 14,15: why the signal of lapis lazuli, very strong, is not detected by Raman spectroscopy; this point should be discussed.

Page 17: the authors can refer to the recent review on the use of cobalt as a colorant (Minerals, 2021, 11, 633), which supports their conclusions.

Author Response

Referee 1

The manuscript is quite well written and can be published after the following issues have been corrected / improved. The main weakness is the absence of a table listing the identified staining agents / phases, their location and which technique proves the presence of the agent / phase as well as the references used for assignment. A comparison of the use of coloring agents identified in other works of the same period (paintings, frescoes) would greatly increase the interest and impact of the manuscript.

We inserted three tables with the main results for each employed technique, discussing the colored areas that we compared within the three walls. Not every hue has been analysed because a large part of the frescoes shows traces of heavy restoration. Sometimes we also found in our measurements traces of these modern restoration materials.  

There are also a few minor issues listed below.

Page 2, line 51 spouses (unclear)

With this term we intended to indicate both Agostino Chigi and his wife. Anyway, we substituted this term.

Page 5, line 135, -1 superscript, same correction in the X-label of Raman figures.

We corrected both the text and the figures

Page 5:  Raman analysis: what is the size of the analysed spot?  

The size of the analysed spot is 100x500μm2

Line 135 fifirst?? Line 136 flfluorescence ?  Page 6, line 142 identiffication, etc. to be corrected

We corrected those mistakes. Still we don’t know how they could happen

Page 6: surface area of the visible reflectance measurement?

The surface area of the visible reflectance measurements is 8 mm2

It is well known that the depth of penetration of x-rays changes with energy (see for example Heritage, 2020, 3, 1302). The measured depth range should be compared with the thickness of the colored fresco layer for the different elements.

We cannot do such comparison because non-invasive analyses do not involve sampling of the surface. It is one of the advantage of such kind of analyses. On the basis of the non-invasive analyses we discussed the main elements from XRF and with the support of Vis-Reflectance we can establish, in some cases, the relative location of the pigments based on the fact that Vis-reflectance individuates only the materials on the surface which are responsible for the “final color”.

Page 7, line 189 : LO Eu peak ? In the absence of factor analysis of groups, such a mention is unnecessary. In addition, in natural hematite, some Fe ions are usually replaced by Ti or Al ions, which makes the description of the mode more complex.  

We removed it

Fig. 4: mark all the XRF peaks; where is the peak of Hg? The measured Raman wave number range makes it difficult to identify HgS; add some evidence / discussion.

The Raman instrument that we used allows for cinnabar identification since its wavenumber range include the intense signal of cinnabar which appears at 342cm-1 and for this reason it was revealed in those areas in which it was present, as the canopy of Roxane’s bed. From XRF sometimes were revealed traces of Hg, too low to be recognized also from Raman. In those cases in which Hg emerges from Raman, it was also revealed by XRF, as we reported in the manuscript.

Page 14,15: why the signal of lapis lazuli, very strong, is not detected by Raman spectroscopy; this point should be discussed.

The lapislazuli signal is not very strong, it is instead rather weak because that area probably underwent  heavy interventions as demonstrated from the other restoration material found in the same zone.

Moreover for identifying lapislazuli with Raman spectroscopy, a laser source of 532nm is needed while our source is operating in the near-infrared and therefore is not suitable to reveal it, and any blue pigments as well. Nevertheless, the few amount of lapislazuli which was left on the surface, suffices to obtain a clear and univocally Vis-Reflectance spectrum as well as to be identified by Infrared spectroscopy.

Page 17: the authors can refer to the recent review on the use of cobalt as a colorant (Minerals, 2021, 11, 633), which supports their conclusions.

We inserted it.

Reviewer 2 Report

In this paper the authors report a very intersting case study. The paper shows a high grade of originality, it is clear and well written. Methods used and results appear significant and interesting.

Very nice the description of the fresco: the reader has actually the impression of visiting the site. Only two typos at lines 135 (..fifirst..), 136 (..fifingerprint..) and 138 (..flfluorescence..)

Author Response

Referee 2

In this paper the authors report a very intersting case study. The paper shows a high grade of originality, it is clear and well written. Methods used and results appear significant and interesting.

Very nice the description of the fresco: the reader has actually the impression of visiting the site. Only two typos at lines 135 (..fifirst..), 136 (..fifingerprint..) and 138 (..flfluorescence..)

We corrected them

Reviewer 3 Report

Please find the review in the attached document

Author Response

Referee 3:

Revision of the article: Things come always in three: Non-invasive investigations of Alexander and Roxane’s Wedding Room in Villa Farnesina.

This paper reports on the application of non-invasive spectroscopic analyses on three walls of Alexander and Roxane’s Wedding Room in Villa Farnesina, two of which attributed to Sodoma, and the third of unknown author. The aim was to identify and compare the painting materials in order to establish a possible common authorship. The use of complementary non-invasive techniques highlights interesting information on the pigments used, corroborating to the hypothesis that an artist closed to Sodoma’s workshop realized the third wall.

I am in favour of publishing this paper in Heritage after the following issues have been addressed:

Materials and methods:

  1. The number of acquisitions for each analysed area is not specified for any of the applied techniques. This information should be added.

With XRF and Vis-Reflectance we performed at least 4 acquisition for each point, while with Raman spectroscopy we did two acquisitions for each point.

Infrared reflectance spectroscopy was used less and only to confirm some hypotheses about pigment as well as to verify if the investigated area had restoration traces.

  1. Further information on Reflectance analysis with the spectrometer 151 CM-700d are needed: illumination area (8 or 3 mm^2), Observer (2° or 10°), Illuminant, measurement conditions (SCI/SCE, illumination/acquisition geometry), number of averaged acquisitions.

Illumination area: 8 mm2, Observer: 10°, Illuminant: D65, measurement conditions: SCI (SCI/SCE, illumination/acquisition geometry), number of averaged acquisitions: 5.

  1. Reflection Infrared Spectroscopy: if I understand well, spectral data were averaged on a sampling area of 28 mm^2, which seems quite big for the analysis of small details. How did you address this issue?

The infrared spectroscopy was used as supporting technique in order to confirm some hypotheses about the nature of pigments as well as to verify if the investigated area had restoration traces. For these reasons there was no need to investigate small details. However an area of 28mm2 is not so big if the fresco surface and the size of the colored areas is considered. This could be a problem in investigating artworks made with painting techniques which involve very small lines of color, as for example in pointillism artworks. In our cases the

Results:

  1. To facilitate the reader, a table summarizing the main pigments identified in each coloured area, with relative spectral features, is needed. This would easy also the comparison among the three walls.

We did it. This was a very important advice, so we thank the referee for having suggested it.

  1. Figures:in general, I would avoid overlying the spectra on the RGB images. The latter can be placed side by side with the graph, as in fig. 6. Note that the colour images in figure 6 are slightly distorted and should be resized. Moreover, in all graphs reporting 1st and 2nd derivatives the y-axis at zero is missing. The yellow lines are not easy to distinguish on the white background (I would use a darker colour).

We did it.

  1. Line 189: I would specify “longitudinal optical” (LO) Eu mode of hematite.

We removed it upon the suggestion of another referee.

  1. Starting from lines 205-209: When reporting the results, consider adding the number between brackets of each analysed area, as it is reported in the figures.

We did it. This also was a very useful suggestion

  1. Lines 220-233: if I understand well, the main identified component of the purple regions is hematite, as in the red areas. The first derivative of the reflectance highlights other spectral differences, which are responsible for the purple hue. For “purple/grey” areas, these spectral differences are attributed to enamel blue (maybe better known as smalt?) due to the presence of Co revealed by XRF. Is the mixture of enamel blue and iron oxide pigments present in all purple areas or only in the purple/grey ones? If yes, please specify and consider adding some references reporting the use of enamel blue in other wall paintings of the same period.  

The presence of smalt in purple/greyish areas is revealed by XRF but it cannot be inferred by the Vis-reflectance alone, because the spectra recorded in these areas show exactly the same shape of those belonging to the purple areas without cobalt. We added other references about the use of the mixture hematite/smalt.

  1. Fig. 7: add some details for Raman bands attribution to specific molecular vibrations (the same for all Raman results)

We did it.

  1. Line 313: “In orange areas, the most evident and common feature of the three walls is the disappearAnce of Sb from the XRF analysis”: This is not clear: does Sb disappear with the respect to the yellow areas? Please, rephrase.  

We rephrased the sentence.

  1. Discussion: this section is relevant but is not clearly written and needs to be rephrased for a good understanding.

We did it.

Please check the typos throughout the text (especially those in section 2: “flfluorescence”, “fifirst”, “fifingerprint”, “fifinally, “reflflection”, reflflectance”…).

We did it.

I recommend a general improvement of the English form too.

We did it.

Reviewer 4 Report

English is fine. Just need to review the text for some minor misspelling, such as in line 127, “mitigate flfluorescence”, line 135 “The fifirst laser”, line 136 “(called fifingerprint”. There are plenty of this kind of strange typos, so it looks like a word editor software issue.

Otherwise, a well-written and interesting paper.  The authors are well know in the field and the paper shows their extensive knowledge into this sector. 

Author Response

Referee 4

English is fine. Just need to review the text for some minor misspelling, such as in line 127, “mitigate flfluorescence”, line 135 “The fifirst laser”, line 136 “(called fifingerprint”. There are plenty of this kind of strange typos, so it looks like a word editor software issue.

Otherwise, a well-written and interesting paper.  The authors are well know in the field and the paper shows their extensive knowledge into this sector

We corrected them.

Round 2

Reviewer 3 Report

The quality of the manuscript has significantly improved after the revisions. Nevertheless, the language should be revised by a native English-speaker before publication.